# Efficacy of Photobiomodulation in Reducing Symptomatology and Improving the Quality of Life in Patients with Xerostomia and Hyposalivation: A Randomized Controlled Trial

**DOI:** 10.3390/jcm11123414

**Published:** 2022-06-14

**Authors:** Alba Ferrandez-Pujante, Eduardo Pons-Fuster, Pia López-Jornet

**Affiliations:** 1Colaborate Department Stomatology, School of Medicine, Oral Medicine University of Murcia, 30100 Murcia, Spain; alba_13_1991@hotmail.com; 2Department Anatomy Psicobiology, Faculty of Medicine, Regional Campus of International Excellence ‘Campus Mare Nostrum’, University of Murcia, 30100 Murcia, Spain; eduardo.p.f@um.es; 3Department of Oral Medicine, Faculty of Medicine, Regional Campus of International Excellence ‘Campus Mare Nostrum’, IMIB Instituto Murciano de Investigación Biosanitaria-Arrixaca, University of Murcia, 30100 Murcia, Spain

**Keywords:** Xerostomia, quality of life, sleep disturbances, anxiety-depression, laser therapy

## Abstract

Objectives: To evaluate the efficacy of photobiomodulation (PBM) treatment in patients with xerostomia and hyposalivation and assess their quality of life over a one year of follow-up. Material and methods: A single-blind randomized controlled trial. A total of 60 patients with xerostomia were included (30 PBM cases and 30 placebo controls). Photobiomodulation was performed with a diode laser (810 nm, 6 J/cm^2^), while the controls underwent simulated treatment. One weekly session was carried out for a total of 6 weeks (total six sessions). The study parameters were recorded at baseline, after three and six weeks, and at one year post-treatment. Xerostomia was assessed using a salivary flow visual analog scale (VAS) and the Xerostomia Inventory (XI). The Hospital Anxiety and Depression (HAD) scale, Oral Health Impact Profile (OHIP-14), Pittsburgh Sleep Quality Index (PSQI) and Epworth Sleepiness Scale (ESS) were also administered. Results: The patients subjected to PBM therapy showed a significant improvement of xerostomia based on the drainage test, and of oral quality of life (*p* < 0.001). The depression score of the HAD (HAD-D) and the ESS showed improvement, though without reaching statistical significance (*p* > 0.05). The placebo group showed significant changes in the xerostomia VAS score at 6 weeks (*p* = 0.009), with no variations in any of the other studied parameters (*p* > 0.05). The beneficial effects of the diode laser in the PBM group persisted at one year post-treatment. Conclusions: Photobiomodulation with the diode laser is effective in patients with xerostomia and hyposalivation, and thus should be taken into account as a treatment option.

## 1. Introduction

Saliva is a body fluid that is essential for oral health and functions. Subjective dry mouth sensation is known as xerostomia, and when a decrease in resting whole saliva flow to below 0.1–0.2 mL/min occurs and in stimulated whole saliva flow to below 0.4–0.7 mL/min is objectively confirmed, hyposialia or salivary hyposecretion is diagnosed [1,2,3]. The most common cause of xerostomia is drug use. In fact, 80% of the most frequently prescribed drugs cause a decrease in salivary gland secretion [4,5]. Apart from drug substances, there are many other potential causes of xerostomia, such as Sjögren’s syndrome, radiotherapy, dehydration or certain emotional events, among others [1,2,3,4]. A lack or reduction in salivary secretion is associated with an increased accumulation of microorganisms on the surfaces of the oral cavity, giving rise to caries, periodontal diseases (gingivitis and periodontitis), denture adjustment and instability problems, lesions caused by rubbing or friction, and halitosis. On the other hand, xerostomia can have functional effects in the form of taste alterations, speech difficulties, as well as chewing and swallowing problems—all of which can have a negative impact upon a patient’s quality of life [5,6,7,8,9,10,11].

Treatments for xerostomia should be established on an individualized basis with measures that range from simple strategies to stimulate saliva output (chewing gum or sweets) to more complex protocols [3,4,5,6,7,8,9]. In this regard, the prescription of bethanechol, pilocarpine or cevimeline may prove effective; however, their many side effects, as well as interactions with other drugs and contraindications in certain clinical situations (peptic ulcer, psychiatric disorders, heart disease and lung disorders, etc.), have led to their limited use. On the other hand, the use of gels and other topical formulations requires multiple applications, imparts a short duration of effect and is costly [6]. To date, it has only been possible to manage xerostomia on a palliative basis—the results often being discouraging in terms of effectiveness. This situation has led to the search for therapeutic alternatives such as photobiomodulation (PBM), which has been widely and successfully used in other diseases, but has been studied to a lesser extent in relation to xerostomia [10,11]. Photobiomodulation (PBM), known as Low Level Laser Therapy (LLLT), is the application of laser or LED to beneficially influence the cellular metabolism. It is a non-thermal and safe treatment [12,13,14].

Photobiomodulation can exert positive effects upon growth factors or cytokine release, and therefore may stimulate cell proliferation and differentiation. Specifically, PBM stimulates the enzyme cytochrome C oxidase within the mitochondria, resulting in the activation of cell signaling pathways [12]. The end effects are accelerated cell metabolism and an increased production of ATP as a useful energy source, with improved cell biochemistry and photochemical processes—resulting in improved cell viability and faster wound healing, as well as anti-inflammatory effects [13,14]. The advantages of PBM that are known thus far include improved epithelial cell mitosis, an increase in salivary ducts, and the stimulation of protein synthesis in submandibular glands. Others report an increase in anti-apoptotic protein expressions and intracellular calcium levels, and blood circulation in the salivary glands that lead to the regeneration of salivary glands and improved functionality [13].

Palma et al. evaluated the effect of PBM in patients with head and neck cancer after radiotherapy and concluded that PBM appears to be effective in mitigating salivary hypofunction and in increasing salivary pH [15]. Saleh et al., in 2014, studied the effect of PBM upon severe hyposalivation and xerostomia as sequelae of head and neck radiotherapy. The authors recorded no significant changes in the study parameters in any of the groups following laser treatment [16]. In turn, Loncar Brzak et al. [17] used lasers of different wavelengths (830 nm and 685 nm). These authors recorded a significant improvement in terms of salivary production in the group of patients with hyposalivation. Few studies have been published to date on the application of PBM to the salivary glands [14,18,19], and there is great variability among them due to the very different protocols involved—making comparisons difficult and sometimes even yielding contradictory findings.

The present study was therefore carried out to investigate the efficacy of photobiomodulation (PBM) applied over six consecutive weeks, with one session per week, and its impact upon hyposalivation and the symptoms of xerostomia.

## 2. Material and Methods

### 2.1. Study Design

A single-blind randomized controlled trial study was carried out at the Dental Clinic of the Department of Oral Medicine (Faculty of Medicine and Dentistry, University of Murcia, Murcia, Spain). The study procedures were all performed by the same operator (AFP). The study protocol abided with the principles of the Declaration of Helsinki and was approved by the Ethics Committee of the University of Murcia (Ref.: 1229/2015). The patients were all volunteers, and provided their informed consent prior to inclusion in the study. The clinical trial followed the guidelines established by the Consort Statement (http://www.consortstatement.org/, accessed on 12 June 2022), and was registered at clinicaltrials.gov (accessed on 12 June 2022) (NCT05336981).

### 2.2. Study Population

Individuals over 18 years of age were enrolled on a consecutive basis. The subjects presented continuous dry mouth symptoms for at least three months, with a resting whole saliva flow of ≤0.1 mL/min.

Patients with unstable medical conditions were excluded, as were cancer patients receiving radiotherapy, individuals with hyperthyroidism or epilepsy, patients using drugs that produce photosensitivity, pregnant women, and individuals with skin lesions in the treatment zone.

### 2.3. Application of Laser PBM

The LaserSmile^®^ low power GaAlAs diode laser (2002 Biolase Technology, Irvine, CA, USA) was used at an operating wavelength of 810 nm, with adoption of the due safety measures in all cases. Laser irradiation was carried out bilaterally after cleaning the treatment skin zone. The salivary glands comprised the parotid gland, with a mean size of 6 × 4 cm (approximate stimulation area 24 cm^2^) and the submandibular gland, with a mean size of 4 × 3 cm (approximate stimulation area 12 cm^2^).

Irradiation was applied externally to the parotid gland on a continuous basis at a dose of 6 J/ cm^2^ for 2 min and 24 s (24 cm^2^ × 6 J/cm^2^ = 144 s; total 1 W) and likewise to the submandibular gland at a dose of 6 J/ cm^2^ but for 1 min and 12 s (12 cm^2^ × 6 J/cm^2^ = 72 s; total 1 W), moving the laser very slowly over the gland zone while in contact with the skin. All patients wore protective goggles during the treatment. In the control group, the laser was likewise applied to the skin over the area of the submandibular and parotid glands, using the same protocol as in the active treatment group, though without activating the laser device.

The study procedure was carried out in six sessions over a period of a month and a half, with one session per week.

The sample size was established considering the previous study published by Fidelix et al. [18]. The level of statistical significance was established as *p* < 0.05, with a statistical power of 80%. The calculated sample size for the comparison of the two groups was 60 subjects (at least 30 in each group). The patients were randomized in a 1:1 proportion to either the active treatment or the control group, using a computer-generated algorithm. Patient assignment to one group or another was kept in a sealed envelope that was not opened until the time of treatment. An external investigator performed the randomization. The professional that applied the PBM treatment did not participate in this process.

## 3. Study Variables

Data collection was carried by means of a standardized clinical interview. Demographic data were recorded, such as patient age and gender, as well as smoking, alcohol consumption, and drug use. The following study parameters were considered:Xerostomia visual analog scale (VAS).

This is a visual scale in which the patient scores dry mouth sensation from 0–10 points, where 0 = no xerostomia and 10 = intense xerostomia. The following formula was used to calculate the percentage improvement of dry mouth sensation: Improvement% = initial VAS—final VAS × 100/initial VAS.

Sialometry.

Sialometry is a measure of saliva flow. The drainage method described by Navazesh et al. was used [20]. The patients were instructed not to eat, drink, smoke or perform oral hygiene 60 min prior to saliva collection, which was performed in the morning. A resting whole saliva flow of ≤0.1 mL/min was considered to be pathological.

Following a resting period, we performed a resting whole saliva flow test (WST I) and a stimulated whole saliva flow test (WST II) using 2% citric acid impregnated in a 1 × 17 cm millimeter paper strip through which the saliva flowed upon contact as described by our group elsewhere [21]. Basal resting flow values of <42 mm and stimulated flow values (stimulation during 5 min) of <75 mm were considered to be pathological.

Xerostomia Inventory (XI).

The Xerostomia Inventory (XI) test was used to obtain information about the severity of the xerostomia symptoms, based on the sum of the scores on a scale involving 11 items. The patients were instructed to choose one out of 5 possible answers corresponding to each of the 11 situations: never (score 1), almost never (score 2), sometimes (score 3), quite often (score 4), and very often (score 5)—with a reference period covering the last four weeks. The total score was between 11 and 55, and reflected the intensity of xerostomia (score 11 = very mild or no xerostomia; 55 = severe xerostomia) [22].

Hospital Anxiety-Depression Scale (HAD).

This instrument consisted of two subscales related to anxiety and depression, respectively. With regard to the interpretation of the HAD scores, a score of >10 indicates the probable presence of anxiety or depression; a score of 8–10 indicates anxiety or depression at the limit of significance; and a score of ≤7 indicates the absence of significant anxiety or depression [23].

Oral Health Impact Profile (OHIP-14).

The OHIP questionnaire, in its short form (14-item) version, was used to assess oral quality of life. The instrument comprises 14 items that explore different aspects of oral function and quality of life. The scores ranged from 0–70, with higher scores indicating a poorer oral quality of life [24].

Pittsburgh Sleep Quality Index (PSQI).

The PSQI is a self-administered questionnaire comprising 19 questions related to 7 domains: subjective sleep quality, sleep latency, duration of sleep, regular sleep efficiency, sleep disturbances, use of medication for sleep, and daytime dysfunction. Each domain is scored from 0–3, with 0 = no problem and 3 = serious problem. The sum of the 7 scores in turn yields a global score from 0–21, with a score of over 5 being considered to be pathological [25].

Epworth Sleepiness Scale (ESS).

The Epworth Sleepiness Scale is a Likert type scale used to assess sleepiness during the day, based on 8 items—with higher scores being indicative of greater sleepiness. A score of 0–9 is considered to be normal [26].

During the first (initial), third (21 days) and sixth (45 days) PBM sessions, we collected data corresponding to whole saliva flow and administered the following study questionnaires: Xerostomia VAS, Xerostomia Inventory (XI), Oral Health Impact Profile (OHIP-14), Hospital Anxiety-Depression Scale (HAD), Epworth Sleepiness Scale (ESS) and the Pittsburgh Sleep Quality Index (PSQI).

In order to assess the evolution of the disorder after a period of one year, the 60 patients were asked to return for clinical review to again collect the resting and stimulated whole saliva values and administer the study questionnaires. Interviews were held by telephone by the same investigator in the case of patients being unwilling or unable to attend the clinic, in order to obtain the required information.

## 4. Statistical Analysis

Analysis of variance (ANOVA) was used for the comparison of more than two means, and the Kolmogorov–Smirnov test confirmed normal data distribution. A descriptive study was performed for each variable, with a calculation of the mean and standard deviation (SD). The Student t-test for paired samples was used for the active treatment and control groups, and the Levene test was used to assess the equality of variances. SPSS version 19.0 statistical package (SPSS Inc., Chicago, IL, USA) was used throughout. Statistical significance was considered at *p* < 0.05.

## 5. Results

Data were obtained for a total of 62 patients with xerostomia that met the inclusion and exclusion criteria, and were enrolled in the study on a consecutive basis. However, two of the patients abandoned the study before starting treatment, due to personal reasons, thus leaving a total of 60 patients with xerostomia that were included in the analysis (Figure 1). The causes of xerostomia were due to drug use for 47 patients, while the remaining 13 patients were diagnosed with Sjogren’s syndrome.

Of the 60 patients, 30 belonged to the active treatment group (mean age 65.4 years; 2 men (6.7%) and 28 women (93.3%)). Four subjects were active smokers (13.3%), while 26 (86.7%) were non-smokers. With regard to alcohol consumption, three of the patients in this group (10%) consumed more than one alcoholic drink a day, while 27 (90%) were not regular drinkers of alcohol.

The remaining 30 patients belonged to the control group (mean age 68.4 years; 100% women). Five subjects were active smokers (16.7%), while 25 (83.3%) were non-smokers. With regard to alcohol consumption, all 30 patients in this group (100%) claimed to not consume alcohol on a regular basis.

The patients subjected to active treatment (PBM) showed an improvement of their symptoms, sialometry data, quality of life and PSQI scores after 6 weeks of treatment versus baseline, and these improvements persisted at 1 year of follow-up. The psychological profile as assessed by the HAD and ESS showed no changes at any of the evaluated timepoints.

In relation to the Xerostomia VAS, the patients in the active treatment group showed a very significant change in score at the third PBM session versus baseline of −32.82% (*p* < 0.001). In turn, at the sixth treatment session, the mean VAS score was 3.9 ± 2.3 (standard deviation), again representing a significant change of −32.76% (*p* < 0.001) with respect to the previous session. At the 1 year follow-up after treatment, the mean VAS score (4.1 ± 2.4) showed a slight increase in dry mouth sensation of 5.41% with respect to the score obtained at the sixth PBM session—though the difference was not statistically significant (*p* = 0.834).

With regard to the Xerostomia Inventory (XI), the patients in the active treatment group showed significant improvement (−42.17%) of the dry mouth scores after the six laser treatment sessions (*p* < 0.001). Likewise, at the 1 year follow-up after treatment, a slight improvement was recorded (−4.17%) that failed to reach statistical significance (*p* = 0.176) (Figure 2).

In the active treatment group, the Oral Health Impact Profile (OHIP-14) indicated a significant increase in oral quality of life, with a decrease in the score of 51.86% after the six treatment sessions (*p* < 0.001). Likewise, at the 1 year follow-up after treatment, the mean score was found to be 8.4 ± 7.1, representing an improvement in patient oral quality of life with respect to the sixth treatment session (1.76% decrease in score) that was not statistically significant (*p* = 0.435) (Figure 3).

In the placebo group, the Xerostomia VAS score showed significant improvement (−7.52%) after the six sessions (*p* = 0.009). At the 1 year follow-up after treatment, the mean score was found to be 7.7 ± 1.6, representing a slight and nonsignificant increase in dry mouth sensation of 2.70% with respect to the score at the sixth treatment session (*p* = 0.274). As can be seen in Table 1, the placebo group showed no significant changes in any of the other study variables.

With regard to saliva drainage, the placebo group showed a 7.11% increase in the recorded values between the first and the sixth treatment session, though this increase was not statistically significant (*p* = 0.085) (Figure 4). At the 1 year follow-up, we recorded many dropouts in the placebo group in relation to saliva flow testing, since the patients did not find the administered treatment to be effective.

No adverse effects were recorded in either group.

## 6. Discussion

In the present study, photobiomodulation (PBM) was seen to increase the production of saliva and reduce the symptoms of xerostomia, resulting in an improvement of the clinical conditions of the patient. Photobiomodulation is a safe and noninvasive technique that is well tolerated.

Cafaro et al. [27] evaluated the effect of PBM upon salivary output at three and six months post-treatment in patients with Sjögren’s syndrome and found the technique to induce a significant increase in salivary output. However, Fidelix et al., in 2018, did not find PBM to improve xerostomia or salivary flow in these patients [18]. These discordant results may be explained by the methodological variability between the studies, with a lack of standardized protocols, since the authors used different laser wavelengths, exposure times, number of sessions, doses, energy densities, follow-up periods, etc. [28].

In this regard, the total energy delivered to the target tissues influences the efficacy of PBM. The upper effective energy density limit is about 30 J/cm^2^. Most studies use energy densities of 2–10 J/cm^2^, with the exception of Loncar, who employed settings of approximately 30 J/cm^2^ [29].

Other parameters that also vary between studies include the site of application of the treatment. In this respect, extraoral irradiation over the gland itself is considered to be more effective than exclusively intraoral administration [14]. In previous biostimulation studies of the sublingual glands, a wavelength of 660 nm was the most widely used setting, though a number of authors used longer wavelengths, particularly for the biostimulation of glands at the extraoral level (780–904 nm) [14,29,30]. We used the same wavelength as Campos-Louzeiro et al. in their study published in 2020 (810 nm) [19]. Long wavelengths afford greater in-depth penetration of the tissues, mainly because hemoglobin and melanin exhibit a greater tendency to absorb light at wavelengths below 600 nm [18,31]. On the other hand, PBM’s operating wavelengths within the red spectrum (600–700 nm) have been associated with a greater incidence of beneficial effects in terms of salivary flow. Wavelengths in the range of 660 nm are well absorbed by cytochrome C oxidase in the target tissues, increasing cellular energy production and thus stimulating the salivary ductal and acinar cells [32].

The studies of xerostomia in patients subjected to radiotherapy offer discordant results. In 2017, Palma et al. [15] evaluated the effect of low-level laser light therapy (LLLT) in 29 patients with head and neck cancer following radiotherapy. The patients underwent two laser therapy sessions a week during 3 months (24 sessions in total), and the authors concluded that LLLT appears to be effective in mitigating salivary hypofunction and in elevating salivary pH. In contrast, other authors such as Oton-Leite et al., in 2013 [33], and Simoes et al., in 2010 [34], recorded no significant improvements in salivary composition or flow, xerostomia or patient quality of life, since all the patients showed a worsening of these parameters or developed some degree of mucositis during their cancer treatment. Nevertheless, PBM was reported to preserve salivary pH during radiotherapy. Saleh et al. evaluated the effect of LLLT upon severe hyposalivation and xerostomia as the sequelae of head and neck radiotherapy, and recorded no significant changes in the parameters analyzed [16]. It has been reported that the main factor conditioning the success of salivary gland stimulation with PBM is the residual viability of the glands [35]. If the salivary glands are irreversibly affected by acinar atrophy or tissue fibrosis, PBM stimulation will be unable to improve salivary flow, since the glands lack residual functional capacity [14,16]. In this respect, Heiskanen et al., in 2020 [36], concluded that it is still too early to firmly establish the efficacy of laser treatment for hyposalivation or xerostomia related to the treatment of cancer.

It would be very useful to stratify xerostomia according to its underlying cause when assessing the efficacy of PBM. Relatedly, in elderly patients with xerostomia due to the use of antihypertensive drugs, Zarvos-Varellis et al. [30] found photodynamic therapy to be effective in reducing xerostomia. In turn, Vidovic-Juras et al., in their study in 2010 [37], found the Xerostomia VAS score to improve as a result of laser treatment. In addition, saliva composition was seen to improve, with an increase in salivary immunoglobulin A. Terlevic-Davic et al. [38] used the GaAlAs laser at a wavelength of 830 nm for the treatment of drug-induced hyposalivation, and recorded a significant increase in unstimulated salivary flow, but not in stimulated flow. In our study, both unstimulated (resting) and stimulated salivary flow were seen to increase as a result of PBM. Mention must also be made of the placebo effect of laser on the Xerostomia VAS score, which was seen to decrease significantly among the controls in our study—though there were no improvements in salivary flow or in any of the other study parameters.

The role of the different PBM parameters remains unclear, and in this regard further research is needed in order to establish uniform protocols and to analyze the long-term management of hyposalivation, considering the quality of life of the patients. Lastly, a recent study published by Campos-Louzeiro et al., in 2020 [39], and a systematic review and meta-analysis conducted by Golež et al., in 2021 [14], described the beneficial effects of PBM upon salivary gland function, showing that the technique alleviates xerostomia and hyposalivation.

Very few studies involve long-term patient follow-up. There appears to be agreement that the effects of PBM decrease after a few months [14,29]. In this regard, the need for quality data and prolonged follow-up periods after PBM must be underscored.

Our study has a number of strong points, including the fact that follow-up covered a period of one year, and the methodology used allowed us to reach the main objectives of the study. In terms of the limitations of our study, mention must be made of its single-center design and the fact that no analysis was made of the changes in salivary pH or composition. A wide variety of causes of xerostomia were included (polypharmacy, Sjögren syndrome), a single-blinded study was performed, and the study procedures were all performed by the same operator.

The data on PBM in patients with hyposalivation and xerostomia vary between studies, and the PBM parameters and protocols are very heterogeneous, with a lack of standardization. Nevertheless, most studies published to date describe promising results, suggesting that the incorporation of PBM to the conventional treatment of xerostomia should be considered.

## 7. Conclusions

Photobiomodulation (PBM) with the diode laser is an effective and noninvasive technique in patients with xerostomia, producing an increase in saliva production, reducing the symptoms of dry mouth, and improving oral quality of life. Further randomized and controlled studies involving longer periods of time are needed to confirm the results obtained.

## Figures and Tables

**Figure 1 jcm-11-03414-f001:**
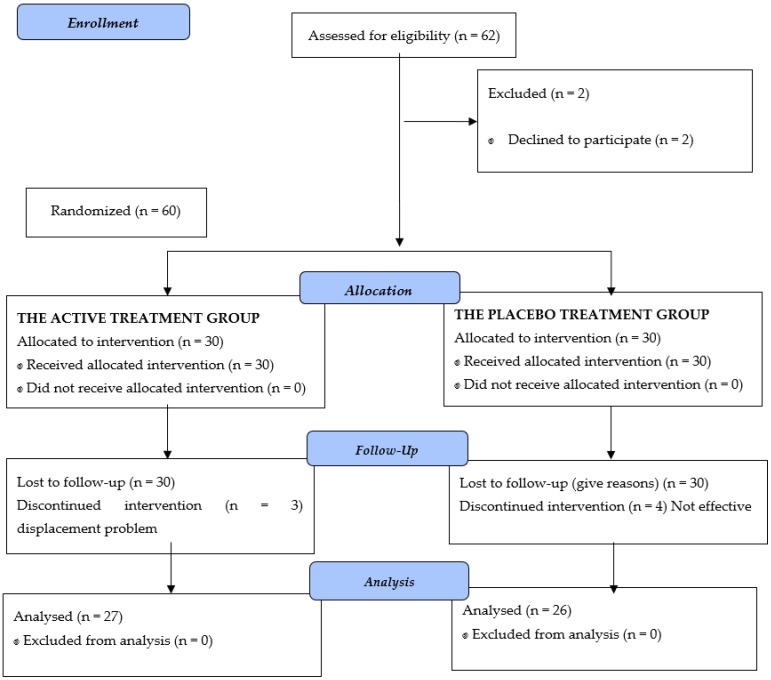
CONSORT 2010 Flow Diagram.

**Figure 2 jcm-11-03414-f002:**
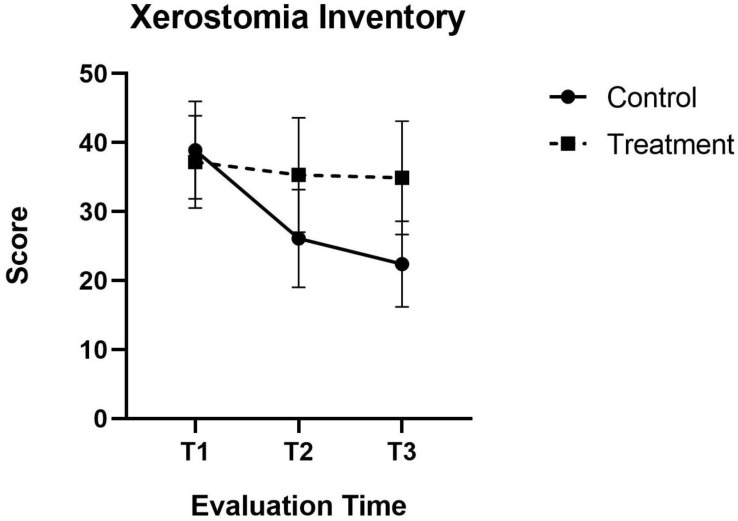
Xerostomia Inventory (XI) Graphs comparing mean (±SD) score at baseline (T1), After 2 weeks of therapy final session (T2) At the end of therapy 6 weeks (T3) experimental laser and control group.

**Figure 3 jcm-11-03414-f003:**
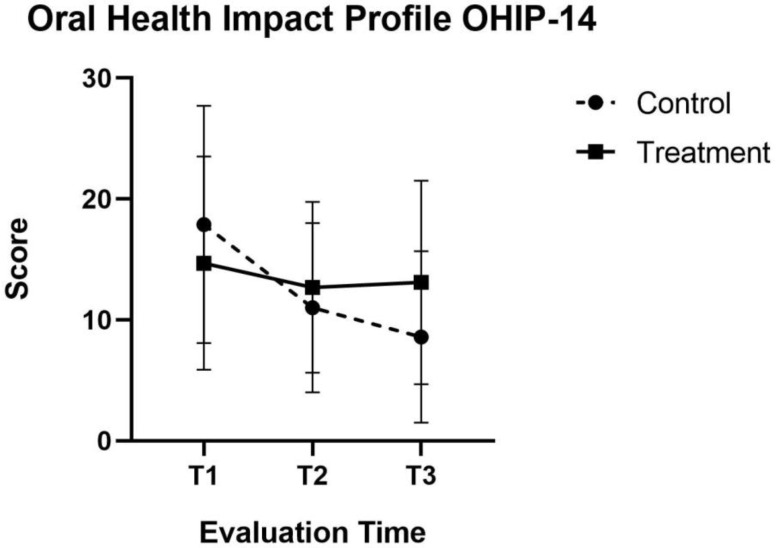
Health Impact Profile (OHIP-14) Graphs comparing mean (±SD) score at baseline (T1), After 2 weeks of therapy final session (T2) At the end of therapy 6 weeks(T3) active laser and control group.

**Figure 4 jcm-11-03414-f004:**
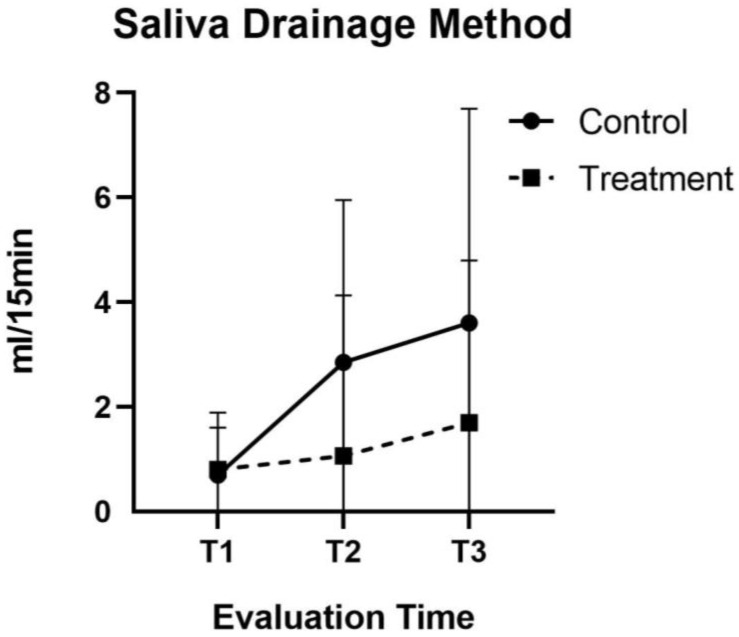
Saliva Drainage method. Graphs comparing mean (±SD) score at baseline (T1), After 2 weeks of therapy final session (T2) At the end of therapy 6 weeks(T3)active laser and control group.

**Table 1 jcm-11-03414-t001:** Efficacy of photobiomodulation (PBM) applied over 6 consecutive weeks. with one session per week. (Treatment group and control group).

Variable	Group	Before Therapy (1)	After 2 Weeks of Therapy	At the End of Therapy (6 Weeks (6))	*p* Value 1 Session a 6 Session	At 1-Year after End of Therapy	*p*-Value 1 Years after the End of Therapy
Xerostomia visual analog scale (VAS)	Treatment	8.6 ± 1.3	5.8 ± 1.9	3.9 ± 2.3	<0.001	4.1 ± 2.4	0.8
	Control	8.2 ± 1.5	7.7 ± 1.7	7.5 ± 1.9	0.009	7.7 ± 1.6	0.2
Drainage method (mL/15 min)	Treatment	0.7 ± 0.9	2.85 ± 3.1	3.6 ± 4.1	<0.001	6.0 ± 8.5	0.5
	Control	0.8 ± 1.09	1.6 ± 3.07	1.7 ± 3.1	0.08	-	-
Whole saliva flow test (basal) mm/5 min)	Treatment	15.6 ± 13.1	39.6 ± 20.8	46.3 ± 27.3	<0.001	47.08 ± 15.9	0.2
	Control	24.1 ± 11.3	31.6 ± 25.5	32.4 ± 26.8	0.1	-	-
whole saliva flow test (stimulated) (mm/5 min)	Treatment	39.8 ± 14.4	72.8 ± 33.2	85.6 ± 61.6	0.005	95.6 ± 56.1	0.02
	Control	47.5 ± 12.9	56.3 ± 53.5	57.1 ± 52.3	0.2	-	-
Xerostomía Inventory	Treatment	38.9 ± 7.06	26.1 ± 7.1	22.5 ± 6.2	<0.001	21.5 ± 8.2	0.1
	Control	37.9 ± 6.7	35.3 ± 8.3	34.9 ± 8.2	0.07		0.4
OHIP14	Treatment	17.9 ± 9.8	11 ± 7	8.6 ± 7.1	<0.001	8.4 ± 7.1	0.4
	Control	14.7 ± 8.8	12.7 ± 7.07	13.1 ± 8.4	0.1	37.1 ± 4.8	0.3
HAD_A	Treatment	9.3 ± 4.4	7.5 ± 3.5	7.1 ± 3.7	0.002	6.9 ± 3.5	0.7
	Control	8.7 ± 4.5	8.2 ± 4.8	8.2 ± 4.8	0.1	15.2 ± 6.8	0.02
HAD-D	Treatment	5.4 ± 4.3	5 ± 4.1	4.7 ± 4.3	0.5	5.1 ± 5.3	0.8
	Control	6.8 ± 5.5	6.8 ± 4.9	7.2 ± 5.1	0.2	10.05 ± 3.8	0.2
PSQ1	Treatment	9.6 ± 3.7	8.2 ± 3.7	7.5 ± 3.8	0.01	8.1 ± 3.7	0.3
	Control	8.6 ± 3.3	8.5 ± 3.02	8.5 ± 3.5	0.8	9.2 ± 2.9	0.7
ESS	Treatment	7.9 ± 5.2	7.3 ± 4.7	7.9 ± 5.3	0.09	6.6 ± 4.3	0.05
	Control	8.1 ± 4.5	8.1 ± 4.8	7.8 ± 4.3	0.547	7.1 ± 3.9	0.3

Note Oral Health Impact Profile (OHIP-14) Hospital Anxiety—Scale (HAD-A); Hospital Depression Scale (HAD-D); Pittsburgh Sleep Quality Index (PSQI); Epworth Sleepiness Scale (ESS).

## Data Availability

The datasets generated during and/or analyzed during the current study are available from the corresponding author on reasonable request.

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
