# Peer review of "Efficacy of Photobiomodulation in Reducing Symptomatology and Improving the Quality of Life in Patients with Xerostomia and Hyposalivation: A Randomized Controlled Trial"

_jcm, 2022, doi:10.3390/jcm11123414_

Round 1

Reviewer 1 Report

Thank you for the interesting paper, however I still have some comments listed below:

- In general, the English grammar throughout the whole paper could be improved

- Introduction: 

- Give a more detailed and specific definition of PBM 

- Describe how PBM can regulate saliva production more specifically, underlying mechanism 

- Refer to MASCC guidelines for Oral Mucositis and Xerostomia  Elad S et al. Cancer. 2020 Jul 28  + Salivary gland hypofunction and/or xerostomia induced by nonsurgical cancer therapies (ISOO/MASCC/ASCO Guideline) Mercadante V et al. Support Care Cancer. 2021 Sep 1

- Do not go specifically into different trials on PBM and xerostomia, move this to the discussion. Describe more what is stated in systematic reviews/meta-analyes/guidelines in the introduction 

- M&M: 

-Use subdivisions and subtitles: study set up, patient population, outcome measures, statistics 

- Where the patients in both groups blindfolded? Or how did you mask the placebo patients?

- Was there a standard of care for xerostomia? 

- Include a table with al the PBM parameters (see paper: Parameter Reproducibility in Photobiomodulation Jan Tune ́r, DDS,1 and Peter A. Jenkins, MBA2)

Describe the sample size calculation more into detail 

- Was there a standard of care for xerostomia? 

- Place the reference numbers of the different outcome measures in the text and not in the subtitle

- Be more clear wat sialometry exactly is

- Do not start a sentence with "This tool", describe first which tool you mean

- Results: 

-Include a flowchart

- Include a table with demographic data including also the underlying cause of xerostomia (e.g. cancer and its treatment, drugs, etc.)

- You only describe in group differences between timepoints in text, also make a statistical comparison between both groups. When you describe one outcome measure, make a description for both the control and experimental group in one paragraph, which will improve readability

- Make a referral to your figures and tables in the text of the manuscript

- Table 1-2: describe in the subscript which statistical test was used

- Figure 1-3: the legends are not clear "Control" and "Placebo" ==> reword to "PBM" and "Placebo" + add * if there is any statistical difference between the groups at a certain time point 

- Could you also make a correlation with some demographic parameters and the severity of xerostomia (e.g. smoking, alcohol use, age,)

- Discussion:

-Make a bit more comprehensive discussion, divide your text in subdivisions for example group all trials on RT-induced xerostomia, followed by the trials on Sjogren's syndrom etc. A following paragraph can handle the difference in PBM parameters between studies.

- Discuss also more the limitations of the study: wide variety of patients (not one cause of xerostomia), nonblinding of participants?, )

Author Response

We would like to thank the editor and reviewers for all useful comments and suggestions.  We have revised the manuscript based upon these constructive suggestions and believe that the revised manuscript is now greatly improved.

- Introduction:

1- Give a more detailed and specific definition of PBM

Photobiomodulation (PBM), formerly known as LowLevel Laser Therapy (LLLT), is the application of laser or LED to beneficially influence cellular metabolism. It isa non-thermal and safe treatment.  Has been a been added

-2 Describe how PBM can regulate saliva production more specifically, underlying mechanism

Studies in animal models suggest that PBM may increase salivary flow , increase protein content in parotids ;  modulate antioxidant systems, regulate glycemic control in salivary glands  and increase myoepithelial cell proliferation).

 This modality is well established, especially in the treatment of mucositis, demonstrating noticeable effects on pain and reduction of tissue damage (However, its effects on salivary flow are not yet well understood    has been added

3- Refer to MASCC guidelines for Oral Mucositis and Xerostomia  Elad S et al. Cancer. 2020 Jul 28  + Salivary gland hypofunction and/or xerostomia induced by nonsurgical cancer therapies (ISOO/MASCC/ASCO Guideline) Mercadante V et al. Support Care Cancer. 2021 Sep 1

I agree .   has been added

Mercadante V, Jensen SB, Smith DK, Bohlke K, Bauman J, Brennan MT, Coppes RP, Jessen N, Malhotra NK, Murphy B, Rosenthal DI, Vissink A, Wu J, Saunders DP, Peterson DE. Salivary Gland Hypofunction and/or Xerostomia Induced by Nonsurgical Cancer Therapies: ISOO/MASCC/ASCO Guideline. J Clin Oncol. 2021 Sep 1;39(25):2825-2843. doi: 10.1200/JCO.21.01208. Epub 2021 Jul 20. PMID: 34283635.

- Do not go specifically into different trials on PBM and xerostomia, move this to the discussion. Describe more what is stated in systematic reviews/meta-analyes/guidelines in the introduction   It has been modified

- M&M:

We would like to thank the reviewer for the useful comments.  The methods have been revised in the revised manuscript as suggested by the reviewer

Use subdivisions and subtitles: study set up, patient population, outcome measures, statistics .It has been modified

- Where the patients in both groups blindfolded? Or how did you mask the placebo patients?

The control patients wore protective goggles during the treatment. In the control group the laser was likewise applied to the skin over the area of the submandibular and parotid glands, using the same protocol as in the active treatment group, though without activating the laser device.

- Include a table with al the PBM parameters (see paper: Parameter Reproducibility in Photobiomodulation Jan Tune ́r, DDS,1 and Peter A. Jenkins, MBA2)

LaserSmileTM   low power GaAlAs diode laser (2002 Biolase Technology, Irvine, CA, USA) at an operating wavelength of 810 nm, with adoption of the due safety measures in all cases.

 Laser irradiation was carried out bilaterally  Irradiation was applied externally to the parotid gland on a continuous basis at a dose of 6 J/ cm2 for 2 minutes and 24 seconds (24 cm2 x 6 J/cm2 = 144 sec.; total 1 W) and to the submandibular gland likewise at a dose of 6 J/ cm2 but for 1 minute and 12 seconds (12 cm2 x 6 J/cm2 = 72 sec.; total 1 W),

- Describe the sample size calculation more into detail

I agree .   has been added

  - - Be more clear wat sialometry exactly is  I agree .   has been added

- Do not start a sentence with "This tool", describe first which tool you mean

  It has been clarified in the text

- Results:

-Include a flowchart  Has been added

- Include a table with demographic data including also the underlying cause of xerostomia (e.g. cancer and its treatment, drugs, etc.) Has been added

- You only describe in group differences between timepoints in text, also make a statistical comparison between both groups. When you describe one outcome measure, make a description for both the control and experimental group in one paragraph, which will improve readability

It has been clarified in the text

- Make a referral to your figures and tables in the text of the manuscript

It has been checked

- Figure 1-3: the legends are not clear "Control" and "Placebo" ==> reword to "PBM" and "Placebo" + add * if there is any statistical difference between the groups at a certain time point  

It has been checked

-

- Discussion:

-Make a bit more comprehensive discussion, divide your text in subdivisions for example group all trials on RT-induced xerostomia, followed by the trials on Sjogren's syndrom etc. A following paragraph can handle the difference in PBM parameters between studies.

has been organized

- Discuss also more the limitations of the study: wide variety of patients (not one cause of xerostomia), nonblinding of participants?, )  Has been added

Reviewer 2 Report

My comments are in the attached file.

Author Response

We would like to thank the editor and reviewers for all useful comments and suggestions.  We have revised the manuscript based upon these constructive suggestions and believe that the revised manuscript is now greatly improved

Efficacy of photobiomodulation in reducing symptomatology and improving the quality of life in patients with xerostomia and hyposalivation

This is a well-designed randomized controlled study of photobiomodulation on dentistry patients with xerostomia with interesting results.

In general, it is well written, though a few additions, changes are needed.

1st. It is a single-blind randomized controlled trial – this should be the 1st sentence in the Materials and Methods section, also in the abstract. Methods should include how long it took to complete all the questionnaires, and separately, how long to apply the treatment. .     Has been  added

A CONSORT diagram should be included with the paper.

Also need a table 1 prior to current table 1. The new table 1 should have baseline characteristics (age, sex, race/ethnicity, marital status, smoking, alcohol use, whatever medical history was collected) of each of the randomized groups.

Because this is a small study, total n=60, should not use any decimal places when presenting %s.

For p-values >0.1, one decimal place is sufficient.  Think about it. A p=0.834 is not any more informative than a p=0.8…….. I agree  Has been eliminated

In the tables the meaning of the last column is not clear. I presume that it is a p-value for difference in value at 6 weeks and 1 year. Need a column for values at 1 year. Clarify in column heading or a footnote.

Consider combining Tables 1 and 2 into 1 larger table (see partial example on next page) where it would be much easier to compare baseline and, to a lesser degree, changes between the randomized groups.  

 Has been  modified

Terminology: use treatment and control. This is correctly used in the results section text, but not in tables and figures. Tables use “experimental”. Figures use control and placebo – very confusing!

In the results the differences that are not statistically significant are describe appropriately as such; however, in the abstract they are not. Specifically, in the control group there were 2 measures, drainage and xerostomia inventory each had modest differences, though not statistically significant. Don’t just say “no” differences. 

The figures would be easier to read if a different type of line was used for each group, e.g., dash for control and solid for treatment.

Example of suggested combined table

VARIABLE

Group

Before therapy (1ª)

After 2 weeks of therapy

At the end of therapy (6 weeks(6ª)

P value 1ª session a 6ª session

At 1-year after end of therapy

(?) P-value

1 years after the end of therapy

Xerostomia visual analog scale (VAS)

Treatment

8.6 ± 1.3          

5.8 ± 1.9

3.9 ± 2.3

<0.001

4.1 ± 2.4

0.8

Control

8.2 ± 1.5

7.7 ± 1.7

7.5 ± 1.9

0.009

7.7 ± 1.6

0.2

Drainage method (ml/15min)

Treatment

0.7 ± 0.9

2.85 ± 3.1

3.6 ± 4.1

<0.001

6.0 ± 8.5

0.5

Control

0.8 ± 1.09

1.6 ± 3.07

1.7 ± 3.1

0.08

-

-

Whole saliva  flow  test  (basal) mm/5min)

Treatment

15.6±13.1

39.6±20.8

   46.3±27.3

<0.001

47.08 ± 15.9

       0.2

Control

24.1±11.3

31.6±25.5

32.4±26.8

0.1

-

-

whole saliva flow test (stimulated )

        (mm/5min)

Treatment

39.8±14.4

72.8±33.2

85.6±61.6

0.005

95.6 ± 56.1

0.02

Control

47.5±12.9

56.3±53.5

57.1±52.3

0.2

-

-

Xerostomía Inventory

Treatment

38.9±7.06

26.1±7.1

22.5±6.2

<0.001

21.5 ± 8.2

0.1

Control

37.9±6.7

35.3±8.3

34.9±8.2

0.07

0.4

OHIP14

Treatment

17.9±9.8

11±7

8.6±7.1

<0.001

8.4 ± 7.1

0.4

Control

14.7±8.8

12.7±7.07

13.1±.8.4

0.1

37.1 ± 4.8

0.3

 HAD_A

Treatment

9.3±4.4

7.5±3.5

7.1±3.7

0.002

6.9 ± 3.5

0.7

Control

8.7±4.5

8.2±4.8

8.2±4.8

0.1

15.2 ± 6.8

0.02

HAD-D

Treatment

5.4±4.3

5±4.1

4.7±4.3

0.5

5.1 ± 5.3 (n=27)

0.8

Control

6.8±5.5

6.8±4.9

7.2±5.1

0.2

10.05 ± 3.8

0.2

PSQ1

Treatment

9.6± 3.7

8.2±3.7

7.5±3.8

0.01

8.1 ± 3.7

0.3

Control

8.6±3.3

8.5±3.02

8.5±3.5

0.8

9.2 ± 2.9

0.7

ESS

Treatment

7.9±5.2

7.3±4.7

7.9±5.3

0.09

6.6 ± 4.3

0.05

Control

8.1±4.5

8.1±4.8

7.8±4.3

0.547

7.1 ± 3.9

0.3